# Secondary Metabolite Profile and Pharmacological Opportunities of Lettuce Plants following Selenium and Sulfur Enhancement

**DOI:** 10.3390/pharmaceutics14112267

**Published:** 2022-10-23

**Authors:** Muna Ali Abdalla, Ibukun Famuyide, Madelien Wooding, Lyndy J. McGaw, Karl H. Mühling

**Affiliations:** 1Institute of Plant Nutrition and Soil Science, Kiel University, Hermann-Rodewald-Str. 2, 24118 Kiel, Germany; 2Phytomedicine Programme, Department of Paraclinical Sciences, Faculty of Veterinary Science, University of Pretoria, Private Bag X04, Onderstepoort 0110, South Africa; 3Department of Chemistry, Natural Sciences 1 Building, University of Pretoria, Private Bag X20, Hatfield 0028, South Africa

**Keywords:** selenium, sulfur, synergistic interaction, NO, antibacterial, cytotoxicity, lettuce, secondary metabolites, UPLC

## Abstract

Selenium (Se) is an essential trace nutrient for humans and animals owing to its role in redox regulation, thyroid hormone control factors, immunity, inflammatory reactions, brain activities, and carbohydrate regulation. It is also important to support muscle development, as well as for reproductive and cardiovascular well-being. Furthermore, sulfur is known to be a healing element, due to the remarkable function of specialized and secondary S-containing compounds. The scope of the current study was to determine the impact of Se and S enrichment on the secondary metabolite accumulation and antibacterial and NO inhibition activities in green and red leaf lettuce (V1 and V2, respectively). The plants were grown in a hydroponic system supplied with different S concentrations (S0: 0, S1: 1 mM and S2: 1.5 mM K_2_SO_4_) via the nutrient solution and foliar-applied varying levels of Se (0, 0.2 and 2.6 µM). Electrospray ionization–quadrupole time-of-flight mass spectrometry (ESI-QTOF/MS) combined with ultra-performance liquid chromatography (UPLC) was used to identify the secondary metabolites in green and red lettuce. The results indicated that extracts of the biofortified lettuce were not cytotoxic to Vero kidney cells at the highest concentration tested of 1 mg/mL. The ESI/MS of the tentatively identified metabolites showed that the response values of 5-*O*-caffeoylquinic acid, cyanidin 3-*O*-galactoside, quercetin 3-*O*-(6′′-acetyl-glucoside) and quercetin 3-*O*-malonylglucoside were induced synergistically under higher Se and S levels in red lettuce plants. The acetone extract of red lettuce had antibacterial activity against *Pseudomonas aeruginosa*, with a minimum inhibitory concentration (MIC) of 0.156 and 0.625 μg/mL under S2/Se1 and S2/Se2 treatments, respectively. As with antibacterial activity, the acetone extract of green (V1) lettuce treated with adequate (S1) and higher S (S2) under Se-limiting conditions showed the ability to inhibit nitric oxide (NO) release from macrophages. NO production by macrophages was inhibited by 50% at respective concentrations of 106.1 ± 2.4 and 101.0 ± 0.6 μg/mL with no toxic effect on the cells, in response to S1 and S2, respectively, under Se-deficient conditions (Se0). Furthermore, the red cultivar (V2) exhibited the same effect as the green cultivar (V1) regarding NO inhibition, with IC_50_ = 113.0 ± 4.2 μg/mL, in response to S1/Se2 treatments. Collectively, the promising NO inhibitory effect and antibacterial activity of red lettuce under the above-mentioned conditions might be attributed to the production of flavonoid glycosides and phenylpropanoic acid esters under the same condition. To the best of our knowledge, this is the first report to show the novel approach of the NO inhibitory effect of Se and S enrichment in food crops, as an indicator for the potential of Se and S as natural anti-inflammatory agents.

## 1. Introduction

Selenium (Se), through various selenoproteins, plays a critical role in the antioxidant defenses of the cell [1]. Selenoproteins contribute to the activation and functions of the cells that regulate the innate and adaptive immune systems [2]. Additionally, there is evidence of the beneficial effects of Se on inflammatory and viral infectious diseases [3]. Furthermore, the selenoenzymes, such as glutathione peroxidases (GPx), have remarkable activity in neutralizing oxidative damage in patients with severe septic shock, and they are involved in inflammatory response regulation. Likewise, a great reduction in serum Se and selenoenzymes, including GPx, was previously found in patients with systemic inflammatory response syndrome and multiple organ dysfunction, characterized by critical sepsis, trauma, acute pancreatitis and burns due to severe oxidative stress [4]. It has been proven that Se supplementation might reduce mortality and ameliorate the clinical outcome [1]. Furthermore, the antibacterial and antifungal activities of Se nanoparticles (SeNPs) have highlighted the medicinal significance of Se [5,6]. For instance, Cremonini et al. [5] found that biogenic SeNPs demonstrated promising antibacterial activity toward clinical isolates of *Pseudomonas aeruginosa*, but had lower antifungal efficacy against *C. albicans*. Sulfur (S) is an important macronutrient for plant growth and development, and it is an essential nutrient for human and animal health. Previous reports indicated that S-containing secondary metabolites exhibited strong anti-inflammatory properties and might have promising potential as future anti-inflammatory candidates [7,8]. Moreover, the antibacterial potential of green lettuce cultivars grown under S treatment towards clinical isolates of the *Staphylococcus aureus* DSM 346 strain has been recently published [9]. Inflammation is a defense mechanism against injury or infection and tissue damage in higher organisms, and it is important to the healing process. It is an immune system’s response mediated by different chemical factors to reduce the initial cause of cell injury [10]. Common symptoms of inflammation are heat, swelling, redness, pain and loss of function. These indicators result from the local release of different immune mediators and the recruitment of immune cells, which direct the response to infection or injury [11,12]. Additionally, the diseases associated with chronic inflammation include cancer, atherosclerosis, chronic inflammatory diseases, diabetes, asthma and autoimmune and degenerative diseases [13]. Nitric oxide (NO) is formed in considerable amounts by the enzyme denoted inducible nitric oxide synthase (iNOS). This enzyme is important for the vasodilation and hypotension observed during inflammation and septic shock. Accordingly, the inhibitors of iNOS might be potential candidates for the management of inflammatory diseases associated with the large production of NO [14]. Hence, NO has long been known for its pro-inflammatory action, associated with essential physiological processes with an effect on the immune system. It would be valuable to determine the inhibitory effect against NO production of lettuce crude extracts following cultivation under different levels of Se and S.

In a previous investigation, secondary metabolites including caffeoyl derivatives, caffeic acid hexose, 5-caffeoylquinic acid, sesquiterpene lactones, luteolin 7-glucuronide and quercetin derivatives were quantified in green and red lettuce by means of HPLC-DAD-MS-based untargeted metabolomics (Figure 1) [9]. Sesquiterpene lactones are well known for their anti-inflammatory, antimicrobial, anticancer, antiviral, immunomodulatory and anti-ulcer properties [15]. Quercetin and kaempferol are among the most recognized metabolites in fruits and vegetables. They have shown strong anti-inflammatory and antioxidant activities in in vitro studies [16], in addition to antihypertensive and cardioprotective properties in animal experiments [17,18]. The absorbed quercetin and kaempferol are consequently metabolized via phase II biotransformation and circulate as methyl, glucuronide and sulfate conjugates [19]. In the current report, the authors would like to propose the following intriguing hypotheses: (1) interaction between Se and S provokes pharmacological properties including antibacterial potential and NO inhibitory effects, especially in red leaf lettuce plants supplied with varied Se and S treatments; (2) secondary metabolites, including different flavonoid glycosides, are induced under Se and S enrichment.

## 2. Materials and Methods

### 2.1. Lettuce Plant Growth and Treatments

The experiment was conducted in a greenhouse where the two leaf lettuce cultivars, known as (V1: Hawking RZ, green multi-leaf lettuce) and (V2: Barlach RZ, red multi-leaf lettuce), were cultivated hydroponically. The seeds were germinated as described by Abdalla et al. [20]. The seedlings were transferred individually into 10 L black containers, arranged with four replicates in a completely randomized design, and kept under standard greenhouse conditions. The basal nutrient solution was made up according to Abdalla et al. [21]. To investigate the influence of Se and S enrichment on the secondary metabolite production, NO inhibitory effect, and antibacterial activity in lettuce plants, Se foliar application was performed once a week for three consecutive weeks, one month after the seedlings were transferred. Additionally, S was applied via the nutrient solution. The Se solution was evenly applied, as described by [21]. The tested treatments were as follows: (1) control (S0Se0: 0 mM K_2_SO_4_ + 0 µM Na_2_SeO_4_); (2) S0Se1: 0 mM K_2_SO_4_ + 0.2 µM Na_2_SeO_4_; (3) S0Se2: 0 mM K_2_SO_4_ + 2.6 µM Na_2_SeO_4_; (4) S1Se0: 1 mM K_2_SO_4_ + 0 µM Na_2_SeO_4_; (5) S1Se1: 1 mM K_2_SO_4_ + 0.2 µM Na_2_SeO_4_; (6) S1Se2: 1 mM K_2_SO_4_ + 2.6 µM Na_2_SeO_4_; (7) S2Se0: 1 mM K_2_SO_4_ + 0 µM Na_2_SeO_4_; (8) S2Se1: 1.5 mM K_2_SO_4_ + 0.2 µM Na_2_SeO_4_; (9) S2Se2: 1.5 mM K_2_SO_4_ + 2.6 µM Na_2_SeO_4_.

The lettuce plants were harvested on the 55th day of the experiment. The lettuce heads were washed, frozen in liquid nitrogen, and placed in a freeze dryer (Gamma1-20, Christ, Osterode am Harz, Germany). The dried samples were ground into fine particles and stored for bioactivity and metabolite analysis.

### 2.2. Extraction

Around 1 g of the dried lettuce heads was extracted with 10 mL of acetone (70%). The crude extract was prepared as described by Famuyide et al., 2019 [22]. A stock solution of 10 mg/mL in 70% acetone was kept for further bioactivity assays.

### 2.3. UPLC-MS Analyses

Dried crude extracts were prepared for UPLC-MS by dissolving 5 mg of the extract in 1 mL MeOH:H_2_O (50:50, *v/v*). The extract was diluted ten times prior to injecting 5 µL on the LC column. Reversed-phase stepwise separation was performed based on a gradient elution scheme starting with 97% H_2_O (0.1% formic acid) to 100% methanol (0.1% formic acid). The gradient started with an isocratic flow, followed by a linear increase to 100% MeOH; subsequently, the column was washed for 2 min, followed by conditioning and re-establishing of initial conditions. A Phenomenex Luna^®^ Omega 1.6 µm EVO C18 100 Å (2.1 mm ID × 100 mm length) column was used. The flow rate (mL/min) was set at 0.3 for the entire run, giving a total run time of 20 min, while the column temperature was kept constant at 50 °C. The positive and negative ion mode mass spectra were obtained in separate chromatographic runs. Thereby, compound separation and identification were performed using a Waters^®^ Synapt G2 high-resolution mass spectrometry (HDMS) system (Waters Inc., Milford, MA, USA). The system comprises a Waters Acquity UPLC^®^ system hyphenated to the QTOF instrument. Data acquisition and processing were performed as previously described [23]. Concerning quantitative data-independent acquisition (DIA), it was performed by using two simultaneous acquisition functions with low and high collision energy (MS^E^ approach) with the aid of a QTOF instrument. Mass spectral scans and the mass of the molecules (*m*/*z*) were collected and recorded as described previously [24].

### 2.4. Nitric Oxide (NO) Production Inhibition Assay in RAW 264.7 Murine Macrophages

#### 2.4.1. Cell Preparation

Macrophages were cultured in 75 cm^2^ flasks in Dulbecco’s Modified Eagle’s Medium (DMEM; Lonza, South Africa) containing L-glutamine supplemented with 10% fetal bovine serum (FBS; Gibco, Sigma-Aldrich, Kempton park, South Africa) and 1% penicillin/streptomycin/fungizone (PSF; Sigma-Aldrich, South Africa) at 37 °C with 5% CO_2_. Subsequently, cells were seeded in 96-well microtiter plates at 4 × 10^5^ cells/mL and incubated overnight at 37 °C with 5% CO_2_ to allow the cells to attach. The cells were consequently activated by the addition of 1 μg/mL of lipopolysaccharide (LPS; Sigma-Aldrich, South Africa), followed by the addition of different concentrations of the extracts. Indomethacin served as a positive control. Cells were incubated for 24 h at 37 °C and 5% CO_2_ [25].

#### 2.4.2. Nitrite Measurement

Following a 24-h incubation period, 100 µL of cell supernatant and 100 µL of Griess (Sigma-Aldrich, South Africa) reagent were added to clean 96-well plates and incubated for 15 min. The absorbance was recorded at 550 nm, as described by Ncube et al., 2021 [26]. The percentage of NO inhibition was calculated as described by [26].

#### 2.4.3. Determination of Cell Viability

To confirm that the NO inhibitory activity observed from the extract was not due to cellular toxicity, a cytotoxicity assay using 3-(4,5-dimethythiazol-2-yl)-2,5-diphenyl tetrazolium bromide (MTT; Inqaba Biotec, Pretoria, South Africa) was performed as described by Mosmann, 1983 [27]. Briefly, after the removal of the supernatant from the RAW macrophages, the cells were washed with phosphate-buffered saline (PBS) and fresh culture medium (200 µL) together with 40 µL of MTT solution [27]. The percentage of cell viability for each sample was calculated by comparing the absorbance in the plant extract-treated wells to the negative control (cells treated with LPS only).

### 2.5. Minimum Inhibitory Concentration (MIC)

The MIC of the crude plant extracts was obtained using a 2-fold serial broth dilution method [28] against the four bacterial strains. Bacterial cultures, namely *Staphylococcus aureus* (ATCC 29213), *Enterococcus faecalis* (ATCC 29212), *Escherichia coli* (ATCC 25922) and *Pseudomonas aeruginosa* (ATCC 27853), were prepared as described by Gado et al., 2021 [29]. The range of concentrations of extracts exposed to the bacteria was 2.5 to 0.02 mg/mL. Gentamicin (Virbac) was used as a positive control, while acetone and broth served as negative and sterility controls, respectively. The diluted bacterial culture was added to the wells of the microtiter plate, which was subsequently incubated at 37 °C overnight. Following incubation, 40 μL p-iodonitro-tetrazolium violet (INT; 0.2 mg/mL) was added to each well and the plates were incubated further at 37 °C for 1 h. The MIC was determined visually as the lowest concentration that led to growth inhibition [29].

### 2.6. Cytotoxicity against Vero Cells

Toxicity assays against monkey kidney (Vero) cells were conducted using the MTT assay, as previously described [20,27,30]. Doxorubicin (Pfizer) and acetone served as positive and negative controls, respectively. After incubation for 48 h, the wells were rinsed with PBS and a fresh medium was added to the cells. Thirty μL of MTT (5 mg/mL dissolved in PBS) was added to each well, followed by a further 4-h incubation period. The absorbance was measured as described previously [20]. Subsequently, the concentration causing 50% inhibition of cell proliferation (CC_50_) was calculated.

### 2.7. Statistical Analysis

The results of ESI/MS responses in positive and negative modes of the selected tentatively identified compounds were statistically analyzed following three-way ANOVA. Samples were tested in triplicate. Subsequently, significant differences among the means for all of the combinations were determined by Tukey’s HSD test (*p* ≤ 0.05). Means of NO inhibitory activity were analyzed using one-way ANOVA. Significant differences in the mean were determined by Dunnett’s multiple comparisons test (* *p* < 0.05; *** *p* < 0.001, and **** *p* < 0.0001).

## 3. Results

### 3.1. Secondary Metabolite Profile Following Se and S Enrichment

In the current study, lettuce plants were enriched with different levels of Se and S supplementation to investigate the dependence of Se and S interaction on the accumulation of the secondary metabolites and subsequent NO inhibition and antibacterial activities. The three Se concentrations were referred to as Se0, Se1 and Se2, which determined Se-limiting conditions (Se = 0 μM), moderate (Se = 0.2 μM) and higher (Se = 2.6 μM) levels, respectively. For the three S treatments, S starvation was expressed as (S0), while adequate and elevated S fertilization was referred to as S1 and S2, respectively.

The resultant interaction between Se and S on the plant can be antagonistic [31,32], but this interaction might be synergistic in the presence of S fertilization in the nutrient solution when Se is applied via the leaves [21]. The interaction between nutrients can be explained as synergistic when the plant’s response to one nutrient enhances the concentration of another nutrient. Our findings exhibit the positive interaction of Se and S on the secondary metabolite levels, as well as the NO inhibitory effect and antibacterial potential, with increasing levels of both elements.

Secondary metabolites were analyzed by means of UPLC-ESI-QTOF/MS and sixteen compounds were tentatively identified by the comparison of the mass fragmentation arrangements using the Waters UNIFI. ESI/MS in both the positive and negative ion mode was used to tentatively identify 16 metabolites in green and red multi-leaf lettuce grown under varying Se and S treatments (Figure 2 and Table 1).

Concerning the ESI/MS response in the positive and negative modes of the tentatively identified compounds, the responses varied under different Se and S levels. For instance, in the case of 5-*O*-caffeoylquinic acid, the ESI/MS response of this compound was enhanced significantly (*p* ≤ 0.05) with increasing Se levels in green lettuce under S deprivation (Figure 3A). However, the response did not alter significantly in red lettuce under S- and Se-limiting conditions, Se0/S0, and with increasing Se levels (Se1 and Se2) in response to S starvation. Furthermore, the response increased synergistically in red lettuce under elevated Se and S concentrations (Se2/S2) compared to the control (Se0/S2). Additionally, cyanidin 3-*O*-galactoside was only detected in red lettuce (Figure 2C). Regarding cyanidin 3-*O*-galactoside, its response value was higher under Se- and S-limiting conditions; however, no significant changes were detected following lower and higher Se doses under S deficiency. Additionally, an elevated response was detected under higher Se and S conditions (Se2S2) (Figure 3B). Concerning quercetin 3-*O*-(6′′-acetyl-glucoside) and quercetin 3-*O*-malonylglucoside, a similar ESI/MS response pattern was detected in green lettuce. For instance, the ESI/MS response values were significantly increased under S deprivation and different Se (Se0, Se1 and Se2) levels compared to the applications of adequate and higher S (S1 and S2) treatments under all Se varied concentrations (Se0, Se1 and Se2) (Figure 3C,D). A drastic increase was detected in red lettuce for both compounds under elevated S and following all Se concentrations (Se0, Se1 and Se2), as listed in Figure 3, for quercetin 3-*O*-(6′′-acetyl-glucoside) and quercetin 3-*O*-malonylglucoside, respectively, in comparison to the treatments of adequate S and all Se levels (Figure 3). A synergistic elevation in ESI/MS response was also deduced for both compounds under S2Se1 and S2Se2, respectively.

Concerning dicaffeoyltartaric acid, its ESI/MS response in green lettuce was induced significantly with increasing Se levels (Se1 and Se2) under S-deficient conditions. The ESI/MS response value was enhanced synergistically to 4605 ± 33 under higher S and Se levels (S2Se2) in green lettuce, while it increased in red lettuce synergistically to 10,938 ± 22 in response to adequate S and higher Se (S1Se2) (Figure 3E). A drastic increase was deduced in red lettuce under S deprivation following all Se (Se0, Se1 and Se2) levels compared to other Se and S treatments in red lettuce.

### 3.2. The Effect of Se and S Interaction on the NO Inhibitory Activity

The anti-inflammatory properties of the lettuce plants grown under varied Se and S treatments were determined by the activities of inducible nitric oxide synthase. 

The RAW 264.7 macrophages are commonly used as a cell line model to assess the production of NO after stimulation with LPS and the effect of plant extracts to inhibit NO by indirect measurements, as reported in several studies [33,34]. 

The results presented in Figure 4A indicate highly statistically significant differences (*p*< 0.001, and *p*< 0.0001) for S1/Se0 and S2/Se0 treatments, respectively in comparison to the control, which exhibited moderate activity at the concentration of 130.5 ± 2.3 μg/mL. Accordingly, the extracts of the green multi-leaf lettuce had moderate NO inhibition, inhibiting up to 50% of NO production at the concentration of 106.1 ± 2.4, and 101.0 ± 0.6 μg/mL in plants treated with adequate and higher S (S1 and S2) levels, respectively, under Se-limiting conditions (Se0). Additionally, a moderate effect was also obtained by the extract of the red lettuce subjected to adequate S (S1) under elevated Se treatments (Se2), as it showed significant differences (*p* < 0.0001) and better activity in comparison to the control (S0Se0; IC_50_ = 191.4 ± 0.4 µg/mL). The plant extract treated with S1/Se2 inhibited up to 50% of NO production at the concentration of 113.0 ± 4.2 μg/mL (Figure 4B). The IC_50_ of the positive control indomethacin was 19.11 µg/mL. However, mild activity was achieved at higher concentrations, where the IC_50_ of the green lettuce extracts treated with adequate S under moderate or elevated Se—S1/Se1 and S1/Se2, respectively (138.4 ± 1.5, and 144.0 ± 2.4 μg/mL)—did not show significant differences under S1/Se1 in comparison to the control (S0Se0; 130.5 ± 2.3 µg/mL) (Figure 4A). However, the IC_50_ under S1/Se2 revealed significant differences (*p* < 0.05) in comparison to the control (S0Se0) (Figure 4A). Additionally, mild activity was observed in red lettuce extracts under greater S and Se fertilization Se2S2 (Figure 4B), where the IC_50_ (132.1 ± 3.1 μg/mL) exhibited significant differences (*p* < 0.0001) compared to the control (S0Se0; IC_50_ = 191.4 ± 0.4 µg/mL).

Less than 50% NO production was obtained at higher concentrations of 243.1 ± 2.3 and 322.8 ± 1.3 µg/mL in response to moderate Se under S deprivation or elevated conditions, respectively (S0/Se1, S2/Se1), in green lettuce. The IC_50_ revealed dramatic significant differences (*p* < 0.0001) compared to the control (S0Se0; 130.5 ± 2.3 µg/mL). Significant differences were observed under adequate or elevated S in response to moderate Se (S1/Se1 and S2/Se1) in red lettuce, with IC_50_ of 220.5 ± 3.0 and 244.2 ± 2.2 µg/mL, respectively, compared to the control (S0Se0; IC_50_ = 191.4 ± 0.4 µg/mL) (Figure 4B).

### 3.3. The Effect of Se and S Interaction on the Antibacterial Activity

Antibiotic resistance is a major concern in the medical community, as it is associated with an increased incidence of severe infections, complications and prolonged hospitalization, which may lead to mortality. Urgent action is needed to slow down the development of antibiotic resistance, and a plant extract with remarkable antibacterial properties could be an alternative approach. 

The crude extracts of multi-leaf green (V1) and red (V2) lettuce were treated with three S levels (S0: 0, S1: 1 mM and S2: 1.5 mM) and three Se treatments (Se0: 0 µM, Se1: 0.2 µM and Se2: 2.6 µM) were tested against Gram-positive bacteria, including *E. faecalis* and *S. aureus*, in addition to Gram-negative bacteria, *E. coli* and *P. aeruginosa.*

The extracts were prepared with 70% acetone. The bacterial culture consequently decreased the acetone concentration initially at 25% in the first well and, subsequently, with a two-fold serial dilution in the following wells. Studies have indicated that microbial growth will not be inhibited by acetone at these concentrations [22]. Negative controls had no effect on the bacteria at the concentrations of acetone tested.

Multi-leaf red lettuce plants grown under higher S and low Se levels (V2S2/Se1) exhibited strong antibacterial activity against *P. aeruginosa* (MIC = 0.156 mg/mL), whereas higher S and Se treatment showed promising activity (MIC = 0.625 mg/mL) against the same microorganism. Both extracts had moderate antibacterial activity towards *E. coli* and *S. aureus* (MIC = 1.25 mg/mL). The MIC values for the positive control gentamicin against *E. coli*, *E. faecalis*, *S. aureus* and *P. aeruginosa* were 0.0008, 0.002, 0.00013 and 0.0003 mg/mL, respectively. On the other hand, the extracts of the multi-leaf green lettuce revealed weak antibacterial activity (MIC >2.5 mg/mL) following different Se and S treatments. This intriguing result provides new insights into how S and Se enrichment has a beneficial effect in enhancing the antibacterial potential of food crops. A contemporaneous investigation studied the synthesis of S and S-Se nanoparticles loaded on reduced graphene oxide and their antibacterial activity against Gram-positive microorganisms. Among the synthesized nanomaterials, rGO-S/Se demonstrated high antibacterial activity against *S. aureus* and *E. faecalis* (growth inhibition > 90% at 0.2 mg/mL) [35]. 

### 3.4. Cytotoxicity (LC_50_) Assay against Monkey Kidney (Vero) Cells

Extracts of leaf lettuce treated with different Se and S applications were evaluated for cytotoxicity against Vero cells. The Vero (African Green Monkey kidney) cells were used as a model for mammalian cells to test the cytotoxicity of the plant extracts. Vero cells are non-cancerous cells and have been used in numerous studies regarding the cytotoxicity of plant extracts, as previously reported [36,37]. They are commonly used in routine toxicological protocols because they are easy to work with and readily available from commercial cell culture repositories, providing consistent responses to the administration of potentially toxic substances [38]. As IC_50_ values for cytotoxicity can vary quite markedly depending on the cell line employed in the study, it has been recommended to select a widely used cell line, such as Vero cells, to enable the comparison of results with those of other researchers [39].

The extracts of the leaf lettuce plants (V1 and V2) subjected to varied Se foliar applications (Se0: 0 µM, Se1: 0.2 µM and Se2: 2.6 µM) together with different S concentrations (S0: 0, S1: 1 mM and S2: 1.5 mM) were not cytotoxic at the highest tested concentration (1 mg/mL). There was therefore relatively low or no cytotoxicity against Vero kidney cells. The positive control was doxorubicin, which had an LC_50_ of 0.013 mg/mL.

## 4. Discussion

Se-enriched food crops may be a valuable resource for reducing Se deficiency, which seriously threatens human health. Additionally, it has been suggested that Se biofortification is beneficial for preventing and treating several chronic diseases [40]. Thus, Se and S enrichment of food crops might induce pharmacological properties in plants owing to their interesting crosstalk. For instance, contemporaneous studies showed that the crosstalk between S and Se in Brassicaceae enhanced the chemopreventive properties of *Brassica* crops by inducing the biosynthesis of Se- or/and S-containing natural metabolites [41]. The current study investigated the effect of Se and S enrichment on antibacterial activity, cytotoxicity, and the NO inhibitory effect, in addition to tentatively characterizing the secondary metabolites in multi-leaf green (V1) and red (V2) lettuce using UPLC-ESI-QTOF/MS. The findings of the current investigation suggest that Se and S enrichment enhances the medicinal significance of the lettuce plants, as well as induces the production of beneficial metabolites, including anthocyanin glycosides, flavonoid glycosides and phenylpropanoid acid esters. Importantly, the results showed that all extracts of the green and red lettuce plants are safe in vitro, with relatively low or no cytotoxicity against Vero kidney cells. It would be useful in further studies to test the extracts against a panel of human cell lines to confirm the lack of cytotoxicity.

Among the tentatively identified metabolites, cyanidin 3-*O*-galactoside (Figure 3B) was only detected in the multi-leaf lettuce plants. This is consistent with what was stated in a previous report, where cyanidin 3-*O*-galactoside and kaempferol 3-(6′′-malonylglucoside) were only detected in the red lettuce [9]. Surprisingly, the ESI response of cyanidin 3-*O*-galactoside was enhanced synergistically, under both higher Se and S levels. The ESI/MS responses of quercetin 3-*O*-(6′′-acetyl-glucoside) and quercetin 3-*O*-malonylglucoside showed a similar pattern in both green and red lettuce plants. An elevated response was detected under S deficiency and all Se (S0, Se1 and Se2) levels for both compounds in green lettuce compared to the application of adequate and higher S under all Se levels, where a reduction was obtained in green lettuce for both compounds under the mentioned conditions (Figure 3C,D). A greatly increased, significant (*p* ≤ 0.05) response was found in red lettuce for both compounds under S- and Se-limiting conditions. In line with several contemporary studies, changes in the polyphenolic metabolites under Se enrichment have been reported [42,43,44]. A synergistic increase in ESI response was also deduced for quercetin 3-*O*-(6′′-acetyl-glucoside) and quercetin 3-*O*-malonylglucoside under S2Se2 in red lettuce, which might support the higher activity under higher Se and S conditions. Hence, although Se is not essential for plants, S is classified as an important macronutrient for plant growth and development. Therefore, S deprivation can result in the induction of various stress conditions. Our data agree with the premise that under S starvation, the prevalence of flavonoids of different chemical subclasses will be enhanced [45]. For instance, under S deprivation, the ESI/MS response values of 5-*O*-caffeoylquinic acid, quercetin 3-*O*-(6′′-acetyl-glucoside) and dicaffeoyltartaric acid in green lettuce were significantly (*p* ≤ 0.05) enhanced, while in red lettuce, the ESI/MS response values of quercetin 3-*O*-malonylglucoside and dicaffeoyltartaric acid were dramatically induced under S-limiting conditions. A contemporary study investigating the metabolite profile of lettuce under S-deficient and S-sufficient conditions indicated that metabolites were significantly altered depending on S nutrition [46]. This can also support the crucial role of S in the functional metabolism of plants.

NO is known as a signaling molecule contributing significantly to the pathogenesis of inflammation. It has an inhibitory effect against inflammation in healthy individuals. However, altered NO homeostasis has been detected during septic shock, and this can cause a reduction in the cardiac index and organ perfusion and lead to organ damage [47]. The results in the present study indicate that adequate or increased S supply under Se-deficient conditions increases the NO inhibitory potential in green lettuce, as extracts inhibited up to 50% of NO production at concentrations of 106.1 ± 2.4 and 101.0 ± 0.6 μg/mL, respectively (Figure 4). This can be attributed to the known fact that natural S possesses anti-inflammatory activity [48]. For instance, organosulfur compounds from garlic eliminated the formation of NO and prostaglandin E2, in addition to their suppressive effect of the pro-inflammatory cytokines tumor necrosis factor-α, interleukin-1β and interleukin-6 in lipopolysaccharide-activated macrophages [7]. Additionally, the red lettuce extracts inhibited up to 50% of NO production at the concentration of 113.0 ± 4.2 μg/mL under adequate S and higher Se.

In addition to this, a higher IC_50_ was obtained at 244.2 ± 2.2 and 322.8 ± 1.3 µg/mL in green and red lettuce at a medium Se concentration and elevated S in V1 and adequate S in V2 (V1S2Se1 and V2S1Se1, respectively); these findings suggest that a moderate Se level has a negative impact on NO inhibition activity in the presence of S. In contrast, an increasing Se concentration under adequate or higher S strengthened the activity in both cultivars (Figure 4A,B), which might be due to the synergistic effect between the two elements. Se has an important contribution in terms of maintaining inflammation, along with its association with selenoproteins [49]. It has been stated previously that a high Se level in the body can fight viral diseases [50]. For instance, a direct link between the course of the human immunodeficiency virus (HIV) disease and the body’s Se amount was discovered. The study indicated that elevated Se levels ameliorate the disease complications and help to reduce the spread of HIV type 1 by inducing glutathione peroxidase activity [51].

The crude extracts of multi-leaf green (V1) and red (V2) lettuce plants subjected to varied Se and S levels were tested for antibacterial activity against a range of Gram-positive (*E. faecalis* and *S. aureus*) and Gram-negative bacteria (*E. coli* and *P. aeruginosa*). The findings showed that the red lettuce plants (V2) grown under elevated S and lower Se levels (V2S2/Se1) had the best antibacterial potential towards *P. aeruginosa* (MIC = 0.156 mg/mL). In addition to this, red lettuce plants treated with higher S and Se levels (Se2S2) synergistically induced promising activity (MIC = 0.625 mg/mL) against *P. aeruginosa*. These findings are in line with the current hypothesis that Se and S enrichment in red lettuce can enhance the antibacterial potential. Furthermore, the red lettuce extracts exhibited moderate antibacterial activity against *E. coli* and *S. aureus* (MIC = 1.25 mg/mL). However, the extracts of the green lettuce grown under different Se and S treatments showed weak antibacterial activity (MIC > 2.5 mg/mL) against all tested microorganisms. In our recent report, the extract of the multi-leaf lettuce (V1) treated with S (S = 0.5 mM; MgSO_4_) exhibited a moderate antibacterial potential of 39.06 ± 7.8% at 200 μg/mL concentrations against the clinically relevant *Staphylococcus aureus* DSM 346 strain. Meanwhile, extracts of the red lettuce (V2) plants cultivated under S sufficiency and S deficiency were equally active, suppressing the bacterial growth at 44.34 ± 5.41% and 43.55 ± 7.34%, respectively, at a similar examined test concentration [9]. Red multi-leaf lettuce is especially rich in anthocyanins [52]; among them, cyanidin 3-*O*-galactoside has been quantified in V2 previously [9]. Cyanidin 3-*O*-galactoside is known for its multiple bioactivities, including antimicrobial potential [53,54]. Various anthocyanins along with cyanidin 3-*O*-galactoside were responsible for the antimicrobial activity of European cranberry extracts against a wide range of human pathogenic Gram-negative and Gram-positive bacteria [54]. In this regard, the increased antibacterial activity and the NO inhibitory properties of red lettuce plants, especially under S2/Se2, might be due to the presence of cyanidin 3-*O*-galactoside, along with several bioactive metabolites. For instance, 5-*O*-caffeoylquinic acid, in addition to quercetin 3-O-(6′′-acetyl-glucoside), quercetin 3-O-malonylglucoside and cyanidin 3-*O*-galactoside, showed higher ESI responses and might be responsible for the entire bioactivity in red lettuce under higher S treatment and lower or higher Se conditions. Additionally, previous studies indicated flavonoid glycosides’ pharmacological properties, including anti-inflammatory and antimicrobial activities [55,56]. Furthermore, the in vivo anti-inflammatory capacity of caffeoylquinic acid derivatives has been reported [57]. The anti-inflammatory properties of plant extracts with high cyanidin 3-*O*-galactoside content have also been determined [53]. 

Dicaffeoyltartaric acid (known as chicoric acid), which is also detected in red lettuce under the above-mentioned conditions has long been considered a safe functional food without unacceptable adverse events. It has been widely used in nutritional supplements and medicines due to its remarkable medicinal significance, including anti-inflammatory, antioxidant and anti-aging properties, in addition to its role in regulating glucose and lipid metabolism [58].

## 5. Conclusions

This is the first report to investigate the promising strategy of researching anti-inflammatory food crop agents by testing lettuce plants treated with Se and S for their NO inhibitory activity. The novelty of this study is that higher Se and S levels (Se2S2), as well as adequate S and higher Se concentrations (Se2S1), can synergistically induce the NO inhibitory properties of red lettuce. Importantly, strong antibacterial activity against *P. aeruginosa* was detected in red lettuce grown under higher S and lower Se levels (S2/Se1). 

Notably, the distinct accumulation of tentatively identified metabolites, annotated as 5-*O*-caffeoylquinic acid, cyanidin 3*-O*-galactoside, quercetin 3*-O*-(6′′-acetyl-glucoside) and quercetin 3-*O*-malonylglucoside, in red lettuce might be responsible for the NO inhibition and antibacterial activities; hence, these metabolites demonstrated high ESI/MS response values under the mentioned conditions. The current report supports the idea that Se and S enrichment in lettuce plants has a great impact on secondary metabolites and bioactivity; hence, it remains highly important to improve the nutritional constituents that nourish human health. Moreover, the analyses presented here indicate that S application can promote the NO inhibition potential in green lettuce cultivars, while, in red lettuce, the NO inhibitory activity improved under moderate S and higher Se. Hence, the red lettuce showed better antibacterial activity compared to the green lettuce, so we can conclude that secondary metabolite production in both cultivars is influenced differently under varying S and Se fertilization regimes. In this regard, the comprehensive and quantitative metabolomic profile will be of benefit to interpret the response of the plant’s overall metabolism under Se and S interaction.

## Figures and Tables

**Figure 1 pharmaceutics-14-02267-f001:**
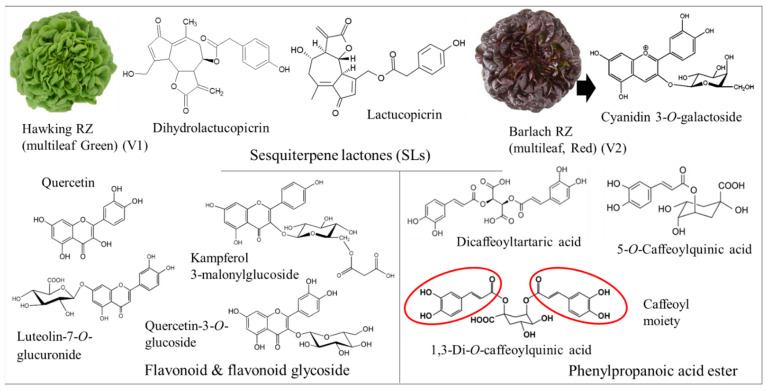
Some secondary metabolites produced by green and red multi-leaf lettuce.

**Figure 2 pharmaceutics-14-02267-f002:**
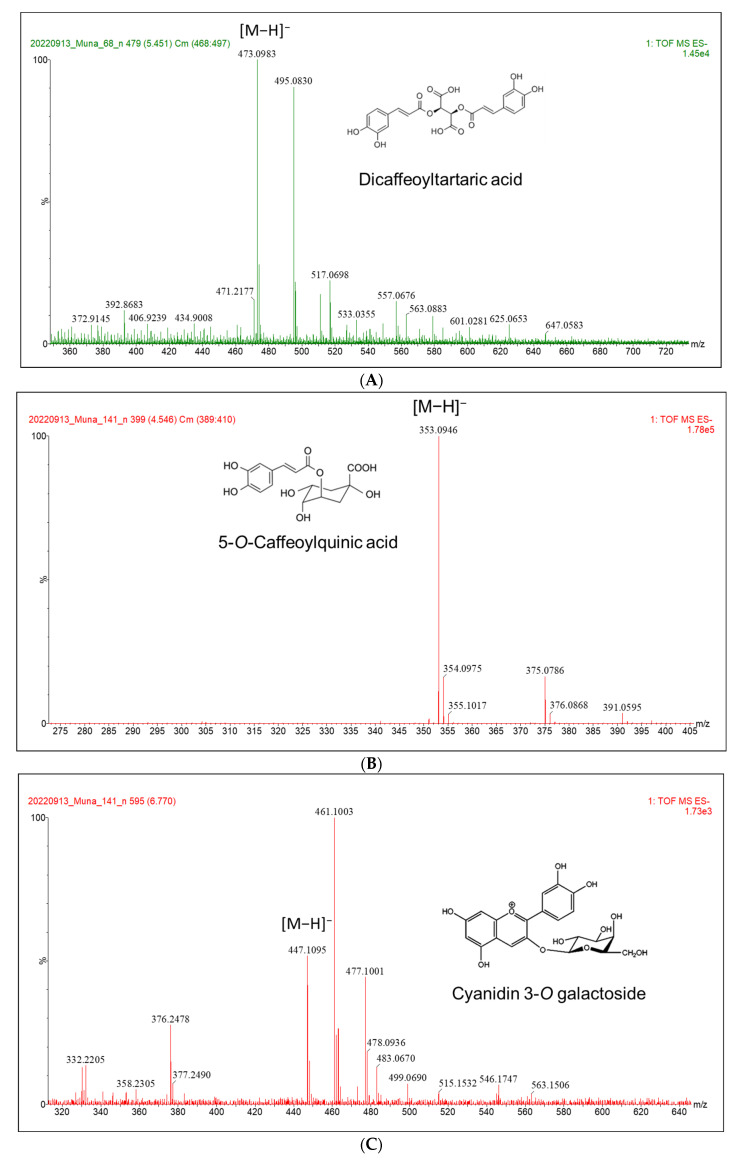
ESI negative mass spectra of (**A**) dicaffeoyltartaric acid (473.0983 *m*/*z;* [M − H]^−^) in green lettuce; (**B**) 5-*O*-caffeoylquinic acid (353.0946 *m*/*z;* [M − H]^−^) and (**C**) cyanidin 3-*O-*galactoside (447.1095 *m*/*z;* [M − H]^−^) in red lettuce. (**D**) Base UPLC peak ion (BPI) chromatograms showing selected metabolites from red lettuce under higher Se and S conditions. (**E**) Overlay of BPI chromatograms from green (green trace) and red lettuce (red trace) under higher Se and S conditions.

**Figure 3 pharmaceutics-14-02267-f003:**
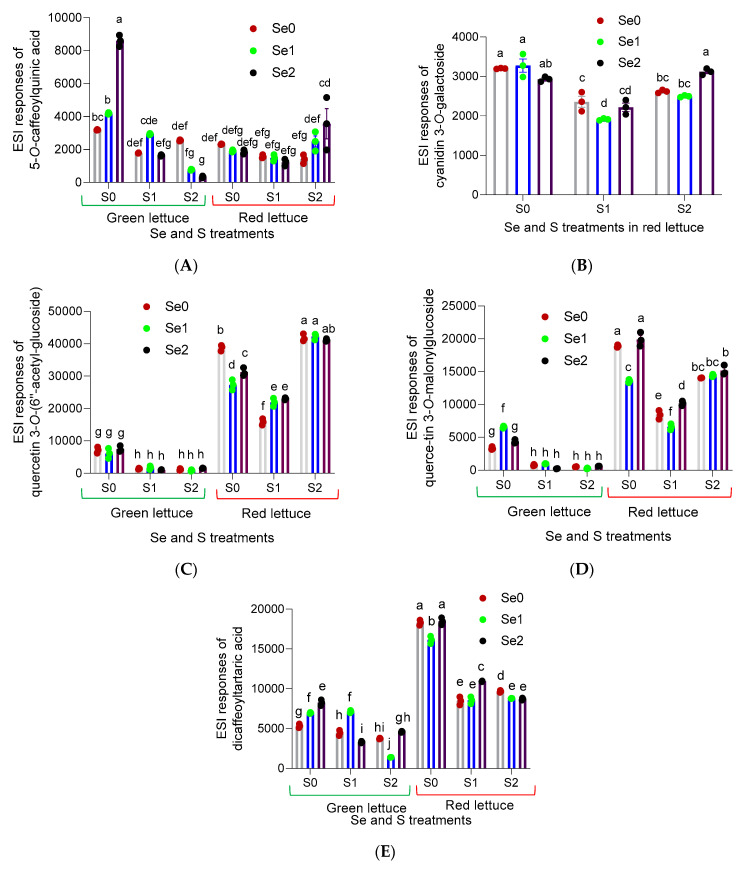
HRMS detector counts of the tentatively identified metabolites in green and red lettuce grown under Se and S enrichment: (**A**) 5-*O*-caffeoylquinic acid; (**B**) cyanidin 3-*O*-galactoside; (**C**) quercetin 3-*O*-(6′′-acetyl-glucoside); (**D**) quercetin 3-*O*-malonylglucoside; (**E**) dicaffeoyltartaric acid. The data presented are the mean ± standard error of the mean (SEM) of three replicates. Different letters show statistically significant differences among all the treatments (*p* ≤ 0.05; Tukey’s test).

**Figure 4 pharmaceutics-14-02267-f004:**
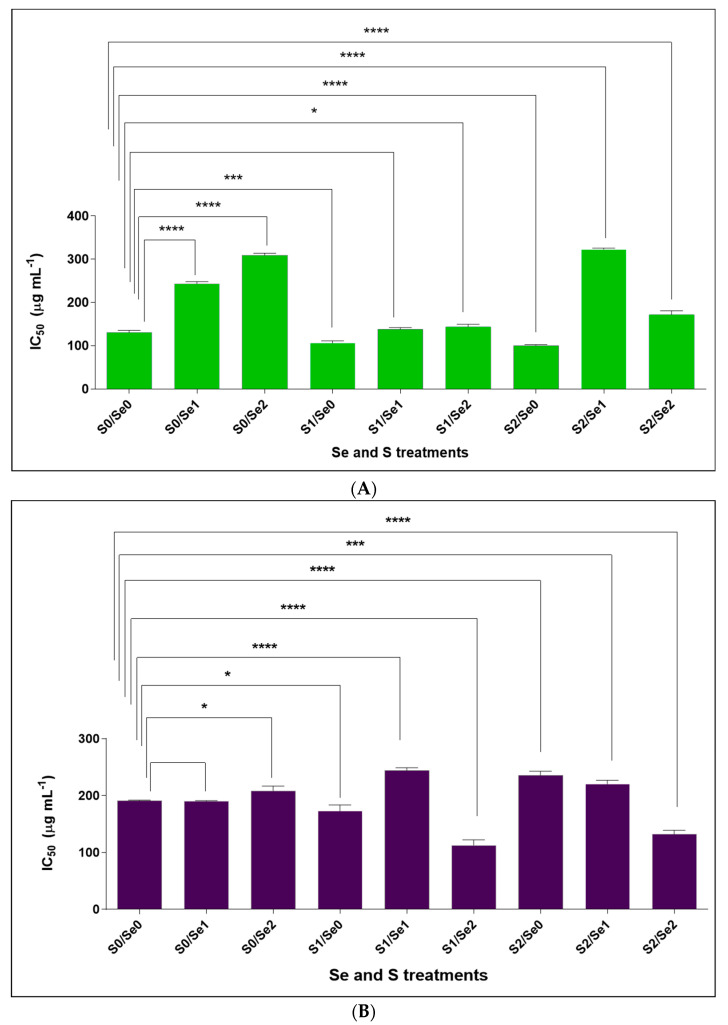
NO inhibitory effect of (**A**) green and (**B**) red lettuce plants grown in a hydroponic system and treated with three S levels (S0: 0, S1: 1 mM and S2: 1.5 mM K_2_SO_4_) and three Se levels (Se0: 0 µM, Se1: 0.2 µM and Se2: 2.6 µM Na_2_SeO_4_). The data presented are the means ± standard error of the means (SEMs) of three replicates. Significant differences in the mean were determined by Dunnett’s multiple comparisons test (** p* < 0.05; **** p* < 0.001, and ******
*p* < 0.0001).

**Table 1 pharmaceutics-14-02267-t001:** Tentative identification of the metabolites in green and red multi-leaf lettuce grown under Se and S treatment using UPLC-HRMS.

Compound Name	Observed *m/z*	Mass Error (mDa)	Observed Retention Time (min)	Detector Counts	Adducts
Luteolin-7,4′-di-*O*-glucoside	611.1626	1.9	6.14	552	[M + H]^+^, [M + Na]^+^, [M + K]^+,^ [M − H]^−^
Quercetin	303.0472	−2.7	8.47	389	[M + H]^+^
5-*O*-Caffeoylquinic acid	377.0826	−1.7	4.52	26,433	[M + Na]^+^, [M + H]^+^, [M − H_2_O]^+^, [M − H]^−^
Cyanidin 3-*O*-galactoside	447.0939	0.6	6.78	3923	[M − H]^−^, [M + H]^+^
Kaempferol 3-(6′′-malonylglucoside)	535.11	1.8	7.71	43	[M + H]^+^
Luteolin-7-*O*-glucuronide	463.0853	−1.8	6.81	8080	[M + H]^+^, [M + Na]^+^, [M − H]^−^
Quercetin 3-*O*-(6′′-acetyl-glucoside)	507.1166	3.2	7.65	43	[M + H]^+^, [M − H]^−^
Quercetin 3-*O*-malonylglucoside	551.1044	1.2	7.2	5378	[M + H]^+^, [M + Na]^+^, [M − H]^−^
Quercetin-3-*O*-glucoside	487.0842	−0.5	7.02	4520	[M + Na]^+^, [M + H]^+^, [M + K]^+^, [M − H]^−^
Dihydrolactucopicrin	413.1585	0.7	4.5	26,433	[M + H]^+^, [M − H]^−^
Lactucopicrin	411.1405	−3.4	8.45	71	[M + H]^+^, [M − H]^−^
Quercetin 3,4′-diglucoside	625.1426	1.5	4.81	518	[M-H]^-^, [M + H]^+^, [M + Na]^+^
Caffeic acid hexoside	341.0885	0.7	3.7	536	[M − H]^−^
Dicaffeoylquininic acid	515.1209	1.4	6.52	6430	[M − H]^−^
Dicaffeoyltartaric acid	473.0723	−0.3	5.62	11,271	[M − H]^−^

## Data Availability

Data are contained within the article.

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
