# Peer review of "Secondary Metabolite Profile and Pharmacological Opportunities of Lettuce Plants following Selenium and Sulfur Enhancement"

_pharmaceutics, 2022, doi:10.3390/pharmaceutics14112267_

Round 1
Reviewer 1 Report
The article ID 1958685 presents the dependence of the influence of S and Se on the biological activity and biosynthesis of active metabolites of lettuce plants. However, before acceptance for publication in Pharmaceutics, the following points should be considered if the Authors decide to revise the work:
1. In Figure 1, the formulas of some polyphenols are incorrect. Please change all sugar rings for the chair conformation. 5-O-caffeoylquinic acid, kaempferol malonylglucoside, was incorrectly represented. The galactose in the formula of Cyanidin 3-O-galactoside is incorrect.
2. In dicaffeoylquinic acid, please identify the position of the caffeic acid residue.
3. Line 89, correctly should be luteolin glucoside derivative or luteolin 7-glucuronide, as it is in Figure 1?
4. In Table 3, instead of Luteolin-7,4'-di-O-β-D-glucopyranoside, it should be Luteolin-7,4'-di-O-glucoside.
5. In Table 3, instead of Quercetin-3-O-β-D-glucopyranoside, it should be Quercetin-3-O-glucoside.
6. In Table 3, the compound Quercetin-3-O-β-D-glucopyranoside in common name is known as isoquercitrin. Why are there two separate compounds with the same name? How do the Authors explain this? Additionally, why are the M+ (m/z) values of these compounds different, 465.0993 and 487.0842 m/z when they refer to compounds of the same molecular weight?
7. In Figure 3, the formulas of 5-O-cavoylquinic acid and Cyanidin 3-O-galactoside are incorrect.
8. In Table 4, the ESI-MS response values shown in positive and negative modes are very unreadable.
9. Please provide two examples of HPLC chromatograms of polyphenol separation in Green and red lettuce 50% methanol extract.
10. In the NO inhibitory test (Figure 2), and antioxidant activity test (Table 1), please show the values of activity parameters for positive standards.
Author Response
Response to the Reviewers’ Comments
Reviewer 1
The article ID 1958685 presents the dependence of the influence of S and Se on the biological activity and biosynthesis of active metabolites of lettuce plants. However, before acceptance for publication in Pharmaceutics, the following points should be considered if the Authors decide to revise the work:
The authors would like to thank the Reviewer for his/her constructive comments and suggestions to improve our manuscript.
Comments and Suggestions for Authors
- In Figure 1, the formulas of some polyphenols are incorrect. Please change all sugar rings for the chair conformation. 5-O-caffeoylquinic acid, kaempferol malonylglucoside, was incorrectly represented. The galactose in the formula of Cyanidin 3-O-galactoside is incorrect.
Response:
Many thanks for this comment. We are highly sorry for the incorrect structures. The structure of 5-O-caffeoylquinic acid, kaempferol malonylglucoside, and cyanidin 3-O-galactoside in Figure 1 were corrected. With respect to the reviewer´s comment, sugar rings were changed to chair conformation.
- In dicaffeoylquinic acid, please identify the position of the caffeic acid residue.
Response:
Many thanks for the reviewer´s comment, caffeoyl moiety was identified in line with the reviewer´s comment.
- Line 89, correctly should be luteolin glucoside derivative or luteolin 7-glucuronide, as it is in Figure 1?
Response:
Many thanks for this comment. Luteolin glucoside derivative was changed to luteolin 7-glucuronide (Line 90).
- In Table 3, instead of Luteolin-7,4'-di-O-β-D-glucopyranoside, it should be Luteolin-7,4'-di-O-glucoside.
Response:
Many thanks for this comment. Luteolin-7,4'-di-O-β-D-glucopyranoside was changed to Luteolin-7,4'-di-O-glucoside (Table 1).
- In Table 3, instead of Quercetin-3-O-β-D-glucopyranoside, it should be Quercetin-3-O-glucoside.
Response:
Many thanks for this comment. Quercetin-3-O-β-D-glucopyranoside was changed to Quercetin-3-O-glucoside (Table 1).
- In Table 3, the compound Quercetin-3-O-β-D-glucopyranoside in common name is known as isoquercitrin. Why are there two separate compounds with the same name? How do the Authors explain this? Additionally, why are the M+ (m/z) values of these compounds different, 465.0993 and 487.0842 m/z when they refer to compounds of the same molecular weight?
Response:
Many thanks for this comment. We are sorry for this mistake, actually, the molecular weight was the same 465.0993[M+H]+ and 487.0842 [M+Na]+ for the two names related to one compound. We changed the name to quercetin-3-O-glucoside as also shown in Figure 1.
- In Figure 3, the formulas of 5-O-cavoylquinic acid and Cyanidin 3-O-galactoside are incorrect.
Response:
Regarding this comment, we corrected both structures in Figure 3.
- In Table 4, the ESI-MS response values shown in positive and negative modes are very unreadable.
Response:
Many thanks for this comment. ESI-MS response values were presented in a figure (Figure 3) instead of a Table.
- Please provide two examples of HPLC chromatograms of polyphenol separation in Green and red lettuce 50% methanol extract.
Response:
Many thanks for this comment. Negative ion mass spectra of different compounds and an overview of extracted ion chromatograms (EIC) of selected metabolites from red lettuce under higher Se and S conditions were provided as in figure 2.
- In the NO inhibitory test (Figure 2), and antioxidant activity test (Table 1), please show the values of activity parameters for positive standards.
Response:
Many thanks for this comment. Regarding NO inhibitory effect, and antibacterial activity the values of activity parameters for positive standards were mentioned (Lines 282-283 and 313-315). All corrections are highlighted in green.

Reviewer 2 Report
The manuscript "Se and S Interrelationship provokes Pharmacological Opportunities and Secondary Metabolites Accumulation in Lettuce Plants" describes the effects of the addition of selenium and sulfur to lettuces in culture and the effects thereof on the metabolites of the crops as well as some biological responses of cells and bacteria against extracts of these crops. The concept of the study is of interest, but there are some major points that need to be answered to make this manuscript more understandable and the design of the study explainable.
Major comments:
1) The complete manuscript needs more structure. The order of the experiments in the materials and methods section should be the same as the order presented in the introduction, the results and the layout of the discussion. It would be more logic to start, after describing the growth of the plants and the extract procedure, with the (bio-)chemical analysis, followed by the microbiological tests and the cell-based assays. This way the observations made on the compounds can be directly linked to the effects seen in the biological experiments, if such links are present.
2) The presentation of the data is in large parts not very good (although the data are probably not bad). Starting with Figure 2, it would be good to know what a mild, moderate and high inhibition is. This could be clarified by including the controls. In the discussion, it should be explained why the responses are not always following clear trends and what the reason might be for a higher IC50 at a medium Se-concentration (green lettuce, S2 and red lettuce S1), compared to the other Se-concentrations. A proper description of the repetitions (technical or biological repeats) and the real interpretation of the Tukey's test is also missing. Table 1 and 2 should be omitted from the manuscript. The data shown in these tables can be easily explained in the text, since there are only very few effects observed. Moreover, it would be good to define a "weak" antimicrobial effect, since the data show no antimicrobial effect. In Table 3, the results of the different treatments cannot be deducted from the data. Table 4 was unfortunately impossible to read due to a formatting error, this should be changed. And finally, Figure 3 also shows very different effects (responses that decrease followed by an increase and visa versa) and this should be explained in the discussion.
3) The discussion is mainly a repetition of the data with very little explanation or interpretation. This should be changed and the parts resembling an introduction or results should be shifted into these parts. The discussion should on the one hand contain the interpretation and on the other hand an (possible) explanation for the observed data. In addition, a number of other issues should be discussed:
3a) why were Vero and RAW 264.7 cells used for the cell culture experiments? These are both animal cell lines (both from another species) and the rationale behind the use of exactly these cells is not given.
3b) The extracts are obtained using acetone. What is the effect of acetone on bacteria and cells? Were any effects observed for the controls and how can potential synergistic effects caused by mixture toxicity (lettuce extract plus acetone) be ruled out? Was the same amount of acetone present in all cultures, independent on the extract concentration?
3c) Is there a possibility to sum up all data (maybe in a figure or table), to see whether there is a correlation between the different end-points that were measured? This would significantly add to the manuscript, since the read-out parameters are relatively different.
Minor comments:
1) The title is difficult to read and does not fully reflect the contents of the paper and it would be better to spell out selenium and sulfur.
2) The results section of the abstract is difficult to understand, since S0/S1/S2 as well as Se0/Se1/Se2 are not explained here. It might be better to state no selenium, low or high concentrations of selenium (and the same for sulfur)
3) Always add a space between a number and a unit (it is not the same throughout the manuscript).
4) All names of bacteria should be written in italics.
5) Line 271 and 272 are not complete correct, since the extracts were not treated with 3 S levels, as is stated in the text.
6) The conclusion reads like a summary and should provide a conclusion on what the data as a whole (and not each test on its own) shows.
Author Response
Reviewer 2
Comments and Suggestions for Authors
The manuscript "Se and S Interrelationship provokes Pharmacological Opportunities and Secondary Metabolites Accumulation in Lettuce Plants" describes the effects of the addition of selenium and sulfur to lettuces in culture and the effects thereof on the metabolites of the crops as well as some biological responses of cells and bacteria against extracts of these crops. The concept of the study is of interest, but there are some major points that need to be answered to make this manuscript more understandable and the design of the study explainable.
Response:
The authors would like to thank the Reviewer for his/her constructive comments and suggestions to improve our manuscript.
Major comments:
1) The complete manuscript needs more structure. The order of the experiments in the materials and methods section should be the same as the order presented in the introduction, the results and the layout of the discussion. It would be more logic to start, after describing the growth of the plants and the extract procedure, with the (bio-)chemical analysis, followed by the microbiological tests and the cell-based assays. This way the observations made on the compounds can be directly linked to the effects seen in the biological experiments, if such links are present.
Response:
Many thanks for this good comment, which really improved the manuscript. We have structured the manuscript in line with the reviewer’s comment. Both materials and methods start with plant extracts, UPLC analysis of the secondary metabolites, followed by biological activities.
2) The presentation of the data is in large parts not very good (although the data are probably not bad). Starting with Figure 2, it would be good to know what a mild, moderate and high inhibition is. This could be clarified by including the controls. In the discussion, it should be explained why the responses are not always following clear trends and what the reason might be for a higher IC50 at a medium Se-concentration (green lettuce, S2 and red lettuce S1), compared to the other Se-concentrations. A proper description of the repetitions (technical or biological repeats) and the real interpretation of the Tukey's test is also missing. Table 1 and 2 should be omitted from the manuscript. The data shown in these tables can be easily explained in the text, since there are only very few effects observed. Moreover, it would be good to define a "weak" antimicrobial effect, since the data show no antimicrobial effect. In Table 3, the results of the different treatments cannot be deducted from the data. Table 4 was unfortunately impossible to read due to a formatting error, this should be changed. And finally, Figure 3 also shows very different effects (responses that decrease followed by an increase and visa versa) and this should be explained in the discussion.
Response:
Many thanks for this comment. Basically, this manuscript deals with the impact of S and Se enrichment on the NO inhibitory and antibacterial activities. The two lettuce cultivars will be differently influenced by Se and S fertilization. A mild, moderate and high inhibition was explained in the manuscript text:
“The results presented in Figure 4 indicate that extracts of the green multi-leaf lettuce had moderate NO inhibition, inhibiting up to 50% of NO production at the concentration of 106.1 ± 2.4, and 101.0 ± 0.6 μg/mL in green lettuce plants treated with adequate S (S1) and higher S (S2) levels, respectively under Se-limiting conditions (Se0). Additionally, the same effect was obtained by the extract of the red lettuce subjected to adequate S (S1) under elevated Se treatments (Se2), as it inhibited up to 50% of NO production at the concentration of 113.0 ± 4.2 μg/mL. The IC50 of the positive control indomethacin was 19.11 µg/mL. However, mild activity was achieved at higher concentrations of 130.5 ± 2.2, 132.1 ± 3.1, 138.4 ± 1.5, and 144.0 ± 2.4 μg/mL in response to different treatments V1/S0Se0, V2/Se2S2, V1/S1Se1 and V1/S1Se2 (Figure 4). Whereas, less than 50% NO production was obtained at higher concentrations of 220.5 ± 3.0, 243.1 ± 2.3, 244.2 ± 2.2, and 322.8 ± 1.3 µg/mL in response to different treatments in green and red lettuce (V2S2Se1, V1S0Se1, V2S1Se1, and V1S2Se1) (Figure 4).” [Results] All corrections are highlighted in green.
what the reason might be for a higher IC50 at a medium Se-concentration (green lettuce, S2 and red lettuce S1), compared to the other Se-concentrations.
Response:
Many thanks for this comment. A response was included in the manuscript text:
In addition to this, a higher IC50 was obtained at 244.2 ± 2.2, and 322.8 ± 1.3 µg/mL in green and red lettuce at a medium Se-concentration and elevated S in V1 and adequate S in V2 (V1S2Se1 and V2S1Se1, respectively), these findings suggest that moderate Se level has a negative impact on NO inhibition activity in the presence of S. In contrast, higher increasing Se concentration under lower or higher S strengthened the activity in both cultivars (Figure 4), which might be due to the synergistic effect between two elements. Se has a great contribution to maintaining inflammation along with its association with selenoproteins. It has been stated previously that a high Se level in the body can fight viral disease. For instance, a direct link between the course of the human immunodeficiency virus (HIV) disease and the body Se amount was discovered. The study indicated that elevated Se level ameliorate the disease complications and help reduce the spread of HIV type 1 by inducing glutathione peroxidase activity (Lines 401-412). All corrections are highlighted in green.
A proper description of the repetitions (technical or biological repeats) and the real interpretation of the Tukey's test is also missing. Table 1 and 2 should be omitted from the manuscript. The data shown in these tables can be easily explained in the text, since there are only very few effects observed.
Response:
Many thanks for this comment. Biological repeats were determined (three replicates) (Lines 198). The real interpretation of Tukey's test (p ≤ 0.05) interpretation was performed. Additionally, Tables 1 and 2 were deleted in response to the reviewer´s comment.
Moreover, it would be good to define a "weak" antimicrobial effect, since the data show no antimicrobial effect
Response:
Many thanks for this comment. "weak" antimicrobial effect, was detected as in lines 316-317.
Table 4 was unfortunately impossible to read due to a formatting error, this should be changed. And finally, Figure 3 also shows very different effects (responses that decrease followed by an increase and visa versa) and this should be explained in the discussion.
Response:
Many thanks for this comment. ESI-MS response values were presented in a figure (Figure 3) instead of a Table. A new figure (Figure 2) was included to show UPLC chromatograms of different compounds in green and red lettuce.
3) The discussion is mainly a repetition of the data with very little explanation or interpretation. This should be changed and the parts resembling an introduction or results should be shifted into these parts. The discussion should on the one hand contain the interpretation and on the other hand an (possible) explanation for the observed data. In addition, a number of other issues should be discussed:
Response:
Many thanks for this comment; based on the reviewer´s comment, we have revised the manuscript, especially the discussion part to highlight our objectives as in lines (358-385, 394-413, 424-451). All corrections are highlighted in green.
3a) why were Vero and RAW 264.7 cells used for the cell culture experiments? These are both animal cell lines (both from another species) and the rationale behind the use of exactly these cells is not given.
Response:
Many thanks for this comment. The Vero (African Green Monkey kidney) cells were used as a model for the mammalian cell to test the cytotoxicity of the plant extracts. Vero cells are known as non-cancerous cells, which is a suitable model that has been used in numerous studies regarding the cytotoxicity of plant extracts as previously reported [36,37]. They are commonly used in routine toxicological protocols, because they are easier to administer and able to increase consistency of response to administration of potentially toxic substances [38]. (Lines 326-331) All corrections are highlighted in green.
The RAW 264.7 macrophage is a frequently used cell line model to assess the production of NO after stimulation with LPS and the effect of plant extracts to inhibit NO by indirect measurements as reported in several studies (Lines 273-275) All corrections are highlighted in green.
Adebayo, S.A.; Dzoyem, J.P.; Shai, L.J.; Eloff, J.N. The anti-inflammatory and antioxidant activity of 25 plant species used traditionally to treat pain in southern African. BMC Complement Altern Med. 2015, 15, 159.
Dzoyem, J.P.; Nkuete, A.H.L.; Ngameni, B.; Eloff, J.N. Anti-inflammatory and anticholinesterase activity of six flavonoids isolated from Polygonum and Dorstenia species. Arch Pharm Res. 2017, 40, 1129–1134.
3b) The extracts are obtained using acetone. What is the effect of acetone on bacteria and cells? Were any effects observed for the controls and how can potential synergistic effects caused by mixture toxicity (lettuce extract plus acetone) be ruled out? Was the same amount of acetone present in all cultures, independent on the extract concentration?
Response:
Many thanks for this comment. The extracts were prepared with 70% acetone. The bacterial culture consequently decreased the acetone concentration initially at 25% in the first well and subsequently, with a 2-fold serial dilution in the following wells. Studies have indicated that the microbial growth will not be inhibited by the acetone at these concentrations (Lines 305-309). Negative controls had no effect on the bacteria at the concentrations of acetone tested.
3c) Is there a possibility to sum up all data (maybe in a figure or table), to see whether there is a correlation between the different end-points that were measured? This would significantly add to the manuscript, since the read-out parameters are relatively different.
Response:
We appreciate the Reviewer’s comment. Significant differences were observed under Se and S treatments, especially among V1 and V2 lettuce cultivars. It remains extremely difficult to discuss the correlation between the different end-points that were measured. Based on the analyses, a much clearer interpretation of the results and discussion was included to clarify the manuscript’s take-home message.
Minor comments:
1) The title is difficult to read and does not fully reflect the contents of the paper and it would be better to spell out selenium and sulfur.
Response:
Regarding this comment, we changed the title as suggested by the reviewer´s comment to "Secondary Metabolite Profile and Pharmacological Opportunities following Selenium and Sulfur enhancement in Lettuce Plants".
2) The results section of the abstract is difficult to understand, since S0/S1/S2 as well as Se0/Se1/Se2 are not explained here. It might be better to state no selenium, low or high concentrations of selenium (and the same for sulfur)
Response:
Many thanks for this comment. Different Se and S treatments were explained as suggested by the reviewer´s comment (Lines 203-209). All corrections are highlighted in green.
3) Always add a space between a number and a unit (it is not the same throughout the manuscript).
Response:
Many thanks for this comment, a space between a number and a unit was performed in the whole manuscript.
4) All names of bacteria should be written in italics.
Response:
Many thanks for this comment, all names of bacteria were written in italics.
5) Line 271 and 272 are not complete correct, since the extracts were not treated with 3 S levels, as is stated in the text.
Response:
Many thanks for this comment. With respect to the reviewer´s comment, the extract was treated with three S levels (S0, S1 and S2) as explained in (Lines 203-209).
6) The conclusion reads like a summary and should provide a conclusion on what the data as a whole (and not each test on its own) shows.
Response:
We appreciate the Reviewer’s comment. The conclusion was rewritten as suggested by the reviewer´s comment:
“Notably, the distinct accumulation of tentatively identified metabolites annotated as 5-O-caffeoylquinic acid, cyanidin 3-O-galactoside, quercetin 3-O-(6''-acetyl-glucoside), and quercetin 3-O-malonylglucoside in red lettuce might be responsible for the NO inhibition and antibacterial activities, hence these metabolites demonstrated high ESI/MS response values under the mentioned conditions. The current report supports the idea that Se and S enrichment in lettuce plants has a great impact on secondary metabolites, and bioactivity. Hence, it remains important to improve the nutritional constituents that nourish human health. Moreover, the analyses presented here indicate that S application can promote NO inhibition potential in the green lettuce cultivar, while in red lettuce the NO inhibitory activity improved under moderate S and higher Se. As the red lettuce showed better antibacterial activity compared to the green ones we can conclude that secondary metabolite production in both cultivars is influenced differently under varying S and Se fertilization regimes. In this regard, the comprehensive and quantitative metabolomic profile will be useful in interpreting the response of the plant’s overall metabolism under Se and S interaction.” [Conclusion]

Round 2
Reviewer 1 Report
The Authors have complied with the reviewer's suggestion, the manuscript has been revised and may be published in Pharmaceutics.
Reviewer 2 Report
The changes made to the manuscript strongly improved the contents and resulted in a study that is much clearer and easier to understand.
One point that it still not fully answered is the statistical test used. In my opinion, the untreated samples should be compared with the treated samples to determine whether there might be a significant effect. The Tukey's test does not do this and showing that 2 groups are similar is not really relevant for this study. I think that it would be good to have a really close look at this or ask a statistician to look at the data.
The choice of cells is also not really well explained. Just because somebody else uses these cells, does not mean that it is the right choice. Even if this study cannot be repeated with different cells at current, it would be of interest for future studies to repeat this data using human cells. This could be added to the discussion.
Small typos are in line 156: microtiter and in line 183 a distance between 37 and °C is missing.